# Hidden Urban Biodiversity: A New Species of the Genus *Scincella* Mittleman, 1950 (Squamata: Scincidae) from Chengdu, Sichuan Province, Southwest China

**DOI:** 10.3390/ani15020232

**Published:** 2025-01-16

**Authors:** Ru-Wan Jia, Zong-Yuan Gao, Di-Hao Wu, Guan-Qi Wang, Gang Liu, Min Liu, Ke Jiang, De-Chun Jiang, Jin-Long Ren, Jia-Tang Li

**Affiliations:** 1CAS Key Laboratory of Mountain Ecological Restoration and Bioresource Utilization, Ecological Restoration and Biodiversity Conservation Key Laboratory of Sichuan Province, Chengdu Institute of Biology, Chinese Academy of Sciences, Chengdu 610213, China; jiarw@cib.ac.cn (R.-W.J.); gaozy@cib.ac.cn (Z.-Y.G.); wudh@cib.ac.cn (D.-H.W.); jiangke@cib.ac.cn (K.J.); jiangdc@cib.ac.cn (D.-C.J.); 2University of Chinese Academy of Sciences, Beijing 100049, China; 3Chengdu Nature Reserve and Wildlife Projection Center, Chengdu 610081, China; wangerjing1997@icloud.com (G.-Q.W.); motion33@outlook.com (G.L.); jing89559@gmail.com (M.L.)

**Keywords:** phylogeny, morphology, *Scincella chengduensis* **sp. nov.**, taxonomy, herpetological diversity

## Abstract

Urban biodiversity is often underestimated, yet new discoveries continue to reveal previously unrecognized species within these environments. This study describes a new species of the genus *Scincella*, *Scincella chengduensis* **sp. nov.**, from the urban and suburban landscapes of Chengdu, Sichuan Province, China. Integrating detailed morphological comparisons and genetic analyses, this species was determined to be distinct from all known skinks in the region. This discovery underscores the role of Chengdu as a biodiversity reservoir, even amidst rapid urbanization. Furthermore, this study highlights the resilience of species in fragmented and human-altered habitats, emphasizing the importance of urban environments for biodiversity research. The discovery of *Scincella chengduensis* **sp. nov.** expands the known diversity of skinks and stresses the urgent need for targeted conservation efforts in urban areas. These findings provide valuable insights for managing urban biodiversity and guiding conservation strategies in cities undergoing rapid development.

## 1. Introduction

The genus *Scincella* Mittleman, 1950 represents a highly diverse group within the family Scincidae Gray, 1825, with a broad geographical distribution across North and Central America, as well as South, East, and Southeast Asia [1,2]. In recent decades, various studies have progressed our understanding of the taxonomy and phylogenetic relationships within this genus [1,3,4]. Despite these efforts, the genus remains taxonomically challenging due to its extensive morphological diversity and the presence of cryptic species complexes.

Recent advances in molecular phylogenetic analyses have shed light on the evolutionary dynamics of *Scincella*, revealing multiple independent radiations and a complex biogeographical history [4,5,6,7,8,9,10,11,12,13,14,15,16]. These analyses have unveiled significant genetic divergence within populations that exhibit morphological similarities, raising critical questions about the true extent of species diversity within the genus. Such findings underscore the importance of integrated approaches that combine morphological and molecular data to refine species delineations and resolve taxonomic ambiguities.

Sichuan Province in China is a recognized hotspot of herpetological diversity, hosting a considerable number of endemic species [17,18,19,20]. The varied topography and climatic conditions of the region create a mosaic of habitats that foster high levels of endemism and speciation. However, the herpetofaunal diversity of Sichuan remains inadequately studied, with extensive areas yet to be systematically surveyed. Preliminary investigations have identified several distinct lineages of *Scincella*, suggesting the presence of undescribed species. Currently, nine species of *Scincella* have been documented in Sichuan, including *S. doriae* (Boulenger, 1887), *S. modesta* (Günther, 1864), *S. monticola* (Schmidt, 1925), *S. potanini* (Günther, 1896), *S. reevesii* (Gray, 1838), *S. schmidti* (Barbour, 1927), *S. tsinlingensis* (Hu and Zhao, 1966), *S. liangshanensis* Jia, Ren & Wu, 2024, and *S. wangyuezhaoi* Jia, Gao, Huang, Ren, Jiang & Li, 2023. The latter two, recently described and endemic to Sichuan, underscore the remarkable biodiversity of the region and the untapped potential for further herpetological exploration [2,14,15,17,18,19,20] (Figure 1).

Chengdu, the capital of Sichuan, is centrally located within the Chengdu Plain and has emerged as a noteworthy area for biodiversity in western China, with recent surveys uncovering remarkable species richness even in its urbanized areas [21,22,23,24,25]. Qing et al. (2013) [26] reconstructed the phylogenetic tree of the genus *Scincella*, but their inclusion of an individual identified as ‘*S. tsinlingensis*’ from the “Sichuan University campus (situated in the urban area of Chengdu)” raises significant questions. Notably, the natural distribution of *S. tsinlingensis* is confined to the Palearctic region, geographically distant from the Chengdu Plain, casting doubt on the validity of this record.

In an ongoing biodiversity survey across Chengdu, as well as in Chongzhou and Dayi, located northwest of the city, previously undocumented populations of *Scincella* were discovered. Morphological analyses revealed that these populations shared similarities with *S. potanini* (Günther, 1896) [15], *S. monticola* (Schmidt, 1925) [27], and *S. liangshanensis* Jia, Ren & Wu, 2024 [28] but displayed distinct characteristics that distinguished them from all described members of the genus *Scincella*. Molecular analyses using both mitochondrial and nuclear DNA sequences confirmed the phylogenetic distinctiveness of this population, supporting its status as a separate species. We herein formally describe this population as a new species within the genus *Scincella*.

## 2. Materials and Methods

**Sampling.** In this study, 90 specimens were examined, including 87 specimens listed in Appendix A and 3 specimens of the newly described species. Due to the limited availability of material, the type series for the new species comprises only 3 specimens. This scarcity is primarily attributed to the species’ restricted distribution and its preference for challenging, inaccessible habitats, making specimen collection difficult. Specimens collected during field surveys were initially preserved in 10% buffered formalin, followed by transfer to 75% ethanol for long-term storage. Tissue samples (liver and muscle) intended for molecular analysis were stored in 95% alcohol at −20 °C to maintain DNA integrity. All specimens were deposited in the Herpetological Museum, Chengdu Institute of Biology (CIB), Chinese Academy of Sciences (CAS), Chengdu, Sichuan Province, China.

**DNA extraction, polymerase chain reaction (PCR), and sequencing.** Genomic DNA was extracted from liver and muscle tissues with a DNA extraction kit (Sangon Biotech, Shanghai, China). Fragments of four mitochondrial genes (16S rRNA (16S), 12S rRNA (12S), cytochrome b (Cyt *b*), and cytochrome oxidase I (*COI*)), alongside one nuclear gene (recombination activating gene 1 (*RAG1*)), were chose based on protocols outlined in Jia et al. (2023, 2024) [14,15] and preliminary experiments. The genes were amplified using the primers followed by Jia et al. (2024) [15]. Gene amplification was performed using primers and PCR conditions described in Jia et al. (2024) [15]. Newly obtained sequences were submitted to GenBank, with accession numbers provided in Table 1. Homologous DNA sequences of voucher specimens from related species were retrieved from GenBank and incorporated into the phylogenetic analyses.

**Phylogenetic analyses.** *Sphenomorphus cryptotis* Darevsky, Orlov, and Cuc, 2004 [5] was chosen as the outgroup to root the tree following Pyron et al. (2013) [29] (Table 1). The four mitochondrial genes and one nuclear gene were sequenced using muscle and liver samples collected from seventeen individuals of the species under study. Additionally, 68 sequences representing 48 individuals from 23 *Scincella* species (including the putative new species) were retrieved from GenBank for comparative analyses, as detailed in Table 1.

Raw nucleotide sequences were manually validated using SeqMan v.7.1.0.44 [30] and subsequently combined with data retrieved from GenBank. Sequence alignment was performed in MEGA X [31] using ClustalW [32] with default parameters to ensure consistency across datasets. The aligned sequences were concatenated with PhyloSuite v.1.2.2 [33]. The optimal partitioning scheme and evolutionary substitution models were determined using PartitionFinder v.2.1.1 [34] with a greedy search algorithm based on the Akaike Information Criterion (AIC).

Phylogenetic trees were constructed using both maximum-likelihood (ML) and Bayesian inference (BI) methods, following the procedures described in Jia et al. (2024) [15]. ML analyses were conducted with RAxML v.8.2.10 [35], while BI analyses were performed in MrBayes v.3.2.6 [36]. Robust node support was considered when Bayesian posterior probability (BPP) was ≥0.95 and ML ultrafast bootstrap support (BS) was ≥70 [37,38]. The phylogenetic trees were visualized with FigTree v.1.4.3 [39]. Uncorrected *p*-distances for 16S were computed using MEGA X [31].

**Morphological analysis.** Morphological analyses followed the methodology outlined in Jia et al. (2024) [15].

Morphological abbreviations and measurement standards followed Jia et al. (2024) [15], including: snout-vent length (**SVL**): distance from tip of snout to posterior edge of vent; tail length (**TaL**): distance from posterior margin of vent to tip of tail; tail width (**TaW**): widest section of tail base; tail depth (**TaD**): ventral to dorsal surface of tail; axilla-groin distance (**AGD**): distance between posterior edge of forelimb insertion and anterior edge of hindlimb insertion; midbody width (**MBW**): measured from lateral surface to opposing lateral edge at midpoint of axillagroin region; midbody depth (**MBD**): measured from ventral surface to dorsal surface at midpoint of axilla-groin region; head length (**HL**): distance from the tip of the snout to the articulation of jaw; maximum head width (**HW**): greatest width between the left and right articulations of jaw; head depth (**HD**): measured from ventral to dorsal surface of head at jaw articulations; eye diameter (**ED**): maximum horizontal diameter of eye; palpebral disc diameter (**PDD**): maximum horizontal diameter of palpebral disc; tympanum diameter (**TD**): ear opening diameter, maximum diameter of ear; eye-narial distance (**END**): from anterior margin of eye to posterior margin of nare; snout length (**SNL**): distance from the tip of the snout to the anterior corner of eye; internasal distance (**IND**): minimum distance between the inner margins of the external nares; interorbital distance (**IOD**): minimum distance between the inner edges of the upper eyelids; forelimb length (**FLL**): measured from forelimb insertion to tip of finger IV or longest digit; hind-limb length (**HLL**): measured from hind-limb insertion to tip of toe IV or longest digit; finger IV length (**F4L**): measured from the most basal part to tip of finger IV; toe IV length (**T4L**): measured from the most basal part to tip of toe IV.

The meristic data included the following: midbody scale-row count (**MBSR**): number of longitudinal scale rows measured around the widest point of midbody; dorsal scale rows between dorsolateral stripes (**DBR**): number of midbody dorsal scale rows between dark dorsolateral stripes; scale rows covered by dorsolateral stripes (**SRB**); enlarged, differentiated nuchal count (**NU**, X pairs or absent); paravertebral scale-row count (**PVSR**): number of scale rows counted between parietals and the just posterior margin of hindlimbs; ventral scale-row count (**VSR**): number of scale rows counted between gulars and precloacals; **gulars**; loreal count (**L**, left/right): number of scale rows counted between the first scale behind the chin-shields and the middle of the forelimb; axilla-groin scale-row count (**AGSR**): number of scale rows counted between posterior edge of forelimb insertion and anterior edge of hind-limb insertion; supralabials (**SL**, left/right); infralabials (**IfL**, left/right); superciliaries (**SC**, left/right); supraoculars (**SO**, left/right); enlarged temporals (**TEM**, left/right); scale-row on dorsal surface of finger and toe (**FTSR**, single or paired); number of enlarged, undivided lamellae beneath finger IV (**F4S**, left/right); number of enlarged, undivided lamellae beneath toe IV (**T4S**, left/right); maxillary tooth (**MT**, left only); lower tooth (**LT**, left only); prefrontals in contact with each other (**PF**, Yes: in contact/No: not in contact/absent); frontoparietals in contact with each other (**FP**, Yes: in contact/No: not in contact/absent); parietals in contact with each other (**P**, Yes: in contact/No: not in contact/absent); **chin-shields**: paired large scales behind mental or postmentals; and limb posture when adpressed, categorized as toes overlapping, in contact, or not in contact with fingers. Dorsal color patterns were also recorded, including upper margin of lateral longitudinal striation wavy or relatively straight (**UMLLS**). Ventral color patterns were assessed, including presence or absence of dark-colored large blotches on ventral (**DLBV**). The raw morphological data for all characters and specimens are presented in Table 2.

Morphological data were derived from published literature [1,2,7,9,12,14,40,41,42,43,44,45,46].

To eliminate the allometric effects of body size, morphometric traits were size corrected by calculating the ratio to SVL. Statistical analyses were performed using Origin 2021 (OriginLab, Northampton, MA, USA). Principal component analysis (PCA) was conducted following the procedures outlined in Jia et al. [15]. Statistical comparisons were conducted between the newly described species and *S. potanini*, *S. monticola*, and *S. liangshanensis* using a Z-score normalized dataset.

## 3. Results

### 3.1. Phylogenetic Analyses

The phylogenetic trees were reconstructed using four mitochondrial genes (12S, 360 bp; 16S, 477 bp; Cyt *b*, 537 bp; *COI*, 623 bp) and one nuclear gene (*RAG1*, 1047 bp) from twenty-four species, resulting in a total alignment length of 3044 bp. Both the ML and BI analyses produced highly consistent topologies, providing strong support for the robustness of the phylogenetic inferences (Figure 2).

Specimens from the Chengdu population formed a distinct monophyletic clade within the genus *Scincella*, displaying substantial genetic divergence from all other congeners. This clade showed the closest relationship to the cluster comprising *S. potanini*, *S. monticola*, and *S. liangshanensis* (BPP = 51; BS = 0.99), although the topology was not fully resolved in the ML analysis. The combination of genetic distances and morphological traits strongly supported the distinctiveness of the Chengdu population.

Uncorrected inter- and intraspecific *p*-distances are presented in Table 3. Results revealed complete genetic identity (0.0%) among specimens from Chengdu (Chongzhou and Dayi), while displaying considerable genetic divergence from other congeners, ranging from 3.0% to 10.4%. The closest genetic similarity was observed with *S. liangshanensis*, whereas the greatest divergence was noted with *S. reevesii* (Gray, 1838) [47].

### 3.2. Morphological Analyses

Morphologically, the specimens from Chengdu were most similar to *S. potanini*, *S. monticola*, and *S. liangshanensis*. However, detailed morphological comparisons (Table 4) showed significant differences between the Chengdu specimens and these species, as well as all other known congeners, particularly in key characters such as VSR, PVSR, AGSR, and DLBV.

The PCA results showed that the first two principal components (PCs) explained 28.1% and 15.3% of the variance, respectively, totaling 43.4%. Scatter plots based on PC1 and PC2 clearly separated the Chengdu specimens from other species with similar morphological traits (Figure 3).

Based on comprehensive morphological and phylogenetic analyses, we concluded that the *Scincella* population from Chengdu, Sichuan, Southwest China, constitutes a distinct new species, described herein.

### 3.3. Taxonomic Account

*Scincella chengduensis* **sp. nov.** Jia, Ren, Jiang, & Li

Figure 4, Figure 5, and Figure 6A

**Holotype.** CIB 118787 (field no. JGS2018016) (Figure 5), adult male, collected from Jiguanshan Forest Park, Chongzhou, Chengdu, Sichuan, China; coordinates 30.77389764° N, 103.22196458° E; elevation 1831 m a.s.l., collected by Ke Jiang, Jin-Long Ren, and Jun Lei on 15 May 2018.

**Paratypes.** CIB 118786, adult female, collected from the same locality as the holotype. CIB 107637, juvenile, collected from Xiling Snow Mountain Scenic Area, Dayi, Chengdu, Sichuan, China; coordinates 30.68313555° N, 103.28359164° E; elevation 1319 m a.s.l.; collected by Yue-Zhao Wang and Yue-Ying Chen on 30 June 2017.

The holotype and two paratypes are preserved in the Herpetological Museum, CIB, CAS.

**Etymology.** The specific epithet is derived from the type locality Chengdu, the capital of Sichuan Province and an important urban center in western China known for its high biodiversity. Reflecting its geographic distribution in Chengdu, the proposed common name is “Chengdu ground skink” in English and “Chéng Dū Huá Xī (成都滑蜥)” in Chinese.

**Diagnosis.** *Scincella chengduensis* **sp. nov.** can be clearly distinguished by a combination of the following unique characters: (1) slender, medium-sized body, snout-vent length 28.4–43.2 mm; (2) infralabials seven, rarely six; (3) supraciliaries six or seven; (4) tympanum deeply recessed without lobules, tympanum diameters equal to or exceeding palpebral disc diameters; (5) midbody scale-row counts 23; (6) dorsal scales smooth, slightly enlarged, paravertebral scale-row counts 57–60, ventral scale-row counts 42–44, gulars 21–22; (7) upper edge of lateral longitudinal stripes relatively straight, four rows of dorsal scales in middle; (8) enlarged, undivided lamellae beneath finger IV 8–9, enlarged, undivided lamellae beneath toe IV 10–12; (9) ventral surface densely covered with dark spots; (10) grayish-brown, irregular dorsal stripes 2–3, black dorsolateral stripes from posterior corner of eye to lateral side of tail.

**Description of holotype.** CIB 118787 (Figure 5): adult male, SVL 37.7 mm; snout short, obtuse; lower eyelid with undivided transparent disc; tympanum recessed, oblique margin prominent; original tail; head elongated, HL 6.8 mm (HL/SVL 0.18), longer than wide, HW 5.1 mm (HW/HL 0.74), lightly flattened, HD 4.2 mm (HD/HL 0.62); neck slender, indistinct from head; scale-row on dorsal surface of fingers and 2nd toe.

**Head:** Snout circular in profile and dorsal views, SNL 2.6 mm, exceeding twice TD (1.1 mm); ear oval; ED 2.0 mm; PDD 0.7 mm, ear opening to snout breadth and palpebral disc ratio 1.58; END 1.6 mm; IND 1.6 mm, IOD 3.1 mm; snout broad, visible dorsally, contacting 1st SL laterally, nasals, and frontonasal posteriorly; MT 20, LT 20; supranasals absent; frontonasal subtrapezoidal, anterior margin forming nearly straight suture (0.6 mm) with rostral, posterior width 1.7 mm, equaling rostral width, exceeding twice its length (0.8 mm), contacting nasals and 1st loreal laterally, slightly touching frontal posteriorly; two prefrontals not in contact, separated medially by frontal, flanked laterally by two loreals, contacting frontal posteriorly; frontal slender, rhombus-shaped, posterior section longer than anterior, contacting 1st and 2nd supraoculars laterally, frontoparietals posteriorly, anterior edge of frontal lightly separating prefrontals medially, posterior edge of frontal lightly overlapping median seam between frontoparietals; two frontoparietals in contact, diamond-shaped, forming butterfly pattern, contacting 2nd–4th supraoculars laterally, interparietal and parietals posteriorly; interparietal small, rhombus-shaped, posterior section longer than anterior, contacting parietals posteriorly, anterior edge of interparietal acute, intruding slightly into median seam between frontoparietals; parietals large, touching posteriorly, narrowly contacting 4th supraocular and posterior supraciliary, broadly contacting anterior secondary temporal laterally and enlarged nuchals posteriorly. Naris circular, located laterally within nasals; nasals contacting 1st SL ventrally, frontonasal dorsally, 1st loreal posteriorly; loreals two, anterior loreal rhomboidal, touching 2nd SL ventrally, frontonasals and prefrontals dorsally, posterior loreal subtrapezoidal, contacting 2nd and 3rd SL ventrally, preocular and upper presubocular posteriorly, prefrontals and anterior supraciliary dorsally; supraciliaries seven, anterior two largest; supraoculars four, first two touching frontal, 2nd to 3rd contacting frontoparietals; lower eyelid with conspicuous transparent disc (window), bordered by small palpebral scales above; supralabials seven, 1st smallest, 5th situated beneath eye window, 6th largest; infralabials seven (left) and six (right), 1st smallest, 5th largest, rectangular or pentagonal; original temporals two, lower larger, sub-rhomboid, contacting 5th and 6th SL ventrally, touching lower secondary temporal posteriorly, anterior primary temporal sub-rhomboid, contacting secondary temporals posteriorly; secondary temporals two, lower smaller, broadly touching anterior, contacting 7th SL ventrally, anterior secondary temporal twice size of lower, contacting parietals dorsally, nuchals posteriorly; nuchals three, bordering posterior parietal edge, enlarged compared to adjacent posterior scales. Mental rounded, contacting 1st IfL laterally, postmental posteriorly; postmental large, contacting 1st and 2nd IfL laterally, 1st chin shield posteriorly; chin shield pairs three, 1st pair broad, contacting medially, touching 2nd–3rd IfL laterally, 2nd pair separated by sub-triangular gular, contacting 3rd–4th IfL laterally, 3rd pair separated medially by three gulars, contacting 5th–6th IfL laterally, three gulars posteriorly; gulars 21.

**Body, limbs, and tail:** Body relatively stout, SVL 37.7 mm; axilla-groin distance relatively long, AGD 23.0 mm (AGD/SVL 0.61); MBW 5.1 mm (MBW/SVL 0.13), MBD 4.6 mm (MBD/SVL 0.12); original tail broken during capture, preserved separately in 75% ethanol, original tail relatively long, TaL 60.0 mm, TaL/SVL 1.59; tail width ≈ height: TaW 4.2 mm (TaW/SVL 0.11), TaD 3.9 mm (TaD/SVL 0.10); forelimbs short, FLL 9.6 mm (FLL/SVL 0.26); hindlimbs longer than forelimbs, HLL 12.2 mm (HLL/SVL 0.32).

Body scales smooth, cycloid, imbricate; dorsal scales larger than lateral scales significantly, larger than ventral scales slightly; anterior flank scales between tympanic region and posterior margin of axilla smaller than adjacent dorsal scales; MBSR 23; PVSR 60; VSR 44; AGSR 58; enlarged preanal scale pair one, median scales overlapping outer scales; dorsal scale rows between dorsolateral stripes 4+2 (1/2); limbs pentadactyl, toes not in contact with fingers when limbs adpressed; digits slender, F4L 2.1 mm, T4L 3.8 mm; F4S 9, T4S 12.

**Coloration in life:** Dorsal surface reddish-brown, marked by two longitudinal stripes formed by contiguous, irregularly shaped black maculations of varying size. Lateral black stripes originate from snout, through supralabials, extend dorsally above eye, continue along flanks above forelimbs and hindlimbs, reaching tail. Axilla-groin stripe black, 1–2 scales wide, with distinct upper boundary. Lateral body surface densely covered with dark spots. Dorsal head brown, with black oval spots. Ventral head and trunk surfaces cream, densely mottled with large, irregular black spots, primarily concentrated along midline. Dorsal limbs brown, ventral surfaces bluish-gray with prominent spotting. Ventral aspect of tail brownish-gray, densely speckled with black spots.

**Coloration in preservative:** Specimens fixed in 10% formalin and preserved in 75% ethanol exhibit a coloration closely resembling that of live animals. However, cream venter changed to bluish-gray and was no longer transparent (Figure 5).

**Variations:** The paratypes (Table 2) are similar to the holotype in most morphometric, meristic, and color traits, with the following variations: (1) PVSR and VSR: 57–60 and 42–44, respectively; (2) gulars: 21–22; (3) AGSR: 52–58; (4) four nuchals on left in CIB 107637; (5) six scales on right in CIB 118786; (6) seven superciliaries in CIB 118786; (7) F4S and T4S: 8–9 and 10–12, respectively; (8) MT and LT: 18–20 and 16–20, respectively. Other minor variations are shown in Table 2.

**Distribution and habitat:** The new species is currently known only from the Jiguanshan Forest Park in Chongzhou, and Dayi, northwestern Chengdu, Sichuan Province, China (Figure 7).

All collected specimens were found in rocky terrain with decaying leaf litter at elevations between 1319 and 1831 m a.s.l. during both dry and wet seasons. The new species is predominantly diurnal, most frequently seen on rocky areas, on leaf-littered cave floors, and in rocky crevices. Sympatric lizard species include *Sphenomorphus indicus* [2,17]. Further research is needed to explore the specifics of their ecological interactions.

**Comparisons:** In terms of pholidosis, *Scincella chengduensis* **sp. nov.** is most similar to *S. potanini*, *S. monticola*, and *S. liangshanensis*, sharing the same number of dorsal scale rows between the dorsolateral stripes (DBR = 4), similar range of enlarged, undivided lamellae beneath toe IV (10–15), and a lack of contact between toes and fingers when limbs are adpressed. However, *Scincella chengduensis* **sp. nov.** can be distinguished from the previous three species by the presence of dark-colored large blotches on the ventral surface. Moreover, *Scincella chengduensis* **sp. nov.** differs from *S. potanini* by having fewer MBSR (23 vs. 24–27), fewer PVSR and VSR (57–60 vs. 62–80 and 42–44 vs. 45–64, respectively), fewer gulars (21–22 vs. 23–25), and fewer AGSR (52–58 vs. 61–82); from *S. monticola* by having a greater number of HL/SVL (0.18–0.20 vs. 0.16–0.17), greater number of FIL/SVL and HLL/SVL (0.21–0.26 vs. 0.13–0.19 and 0.23–0.32 vs. 0.20–0.22, respectively), fewer PVSR and VSR (57–60 vs. 69–73 and 42–44 vs. 45–52, respectively), and fewer AGSR (52–58 vs. 65–74).

*Scincella chengduensis* **sp. nov.** was recovered as the sister species of *S. liangshanensis*, with the *p*-distance between this species pair representing the closest genetic resemblance (3.0%). Nevertheless, *Scincella chengduensis* **sp. nov.** can be readily distinguished from the latter species by having fewer PVSR (57–60 vs. 69–80), fewer VSR + gulars (64–65 vs. 68–82), fewer AGSR (52–58 vs. 60–79), and shorter SVL (28.4–43.2 mm [*n* = 3] vs. 43.1–61.9 mm [*n* = 16]), as well as the aforementioned differences in ventral color pattern (Figure 3 and Figure 6; Table 4).

For other congeners also showing four dorsal scale rows between dorsolateral stripes, *Scincella chengduensis* **sp. nov.** differs from *S. tsinlingensis* (Hu and Zhao, 1966) and *S. huanrenensis* Zhao and Huang, 1982 by having fewer MBSR (23 vs. 26–28 and 25–28, respectively), fewer PVSR (57–60 vs. 60–75 and 66–79, respectively), fewer VSR + gulars (64–65 vs. 83–98 and 75–83, respectively), and fewer T4S (10–12 vs. 13–16) [48,49,50]; from *S. schmidti* (Barbour, 1927) [51] by having a shorter tail, TaL/SVL (1.59 vs. 1.90), fewer MBSR (23 vs. 26), fewer PVSR (57–60 vs. 66), fewer VSR + gulars (64–65 vs. 71), and fewer AGSR (52–58 vs. 60).

For other Chinese congeners displaying six or eight dorsal scale rows between dorsolateral stripes, *Scincella chengduensis* **sp. nov.** differs from *S. reevesii* and *S. barbouri* (Stejneger, 1925) by having fewer AGD/SVL (0.55–0.61 vs. 0.61–0.66 and 0.61–0.65, respectively), greater FIL/SVL (0.21–0.26 vs. 0.16–0.19 and 0.17–0.20, respectively), fewer MBSR (23 vs. 26–32 and 26–28, respectively), fewer T4S (10–12 vs. 15–18 and 15–17, respectively), and relatively straight UMLLS (vs. wavy) [1,52]; from *S. doriae* (Boulenger, 1887) by having fewer MBSR (23 vs. 30–32), fewer PVSR (57–60 vs. 66–76), fewer VSR+gulars (64–65 vs. 70–79), fewer T4S (10–12 vs. 15–18), relatively straight UMLLS (vs. wavy), and toes not in contact with fingers when limbs adpressed (vs. overlapping) [3,44,53,54]; from *S. formosensis* (Van Denburgh, 1912) by having fewer HLL/SVL (0.23–0.32 vs. 0.34–0.39), having fewer MBSR (23 vs. 26–28), more PVSR (69–80 vs. 53–65), fewer T4S (10–12 vs. 14–17), and relatively straight UMLLS (vs. wavy) [40,55]; from *S. modesta* (Günther, 1864) by having fewer MBSR (23 vs. 26–28), fewer AGSR (52–58 vs. 58–70), fewer T4S (10–12 vs. 13–15), and relatively straight UMLLS (vs. wavy) [56]; from *S. przewalskii* (Bedriaga, 1912) by having fewer HLL/SVL (0.23–0.32 vs. 0.33), more supraoculars (4 vs. 3) and fewer T4S (10–12 vs. 17) [40]; from *S. wangyuezhaoi* Jia, Gao, Huang, Ren, Jiang & Li, 2023 by having fewer MBSR (23 vs. 27–30), fewer PVSR (57–60 vs. 60–75), fewer VSR + gulars (64–65 vs. 73–86), and fewer T4S (10–12 vs. 14–15) [14].

*Scincella chengduensis* **sp. nov.** can be clearly differentiated from its Asian congeners based on key morphological traits. Notably, the new species contains fewer MBSR compared to most other congeners (23 vs. 24–36), except for *S. apraefrontalis* Nguyen, Nguyen, Böhme & Ziegler, 2010 (23 vs. 18) [11]. Further distinctions include whether toes are in contact with or overlap fingers when limbs are adpressed. For species in which toes overlap with fingers when limbs are adpressed, *Scincella chengduensis* **sp. nov.** can be distinguished by having fewer PVSR (57–60 vs. 63–74), fewer DBR (4 vs. 6–10), and fewer T4S (10–12 vs. 16–22). This pattern is consistent across species such as *S. badenensis* Nguyen, Nguyen, Nguyen & Murphy, 2019, *S. melanosticta* (Boulenger, 1887), *S. nigrofasciata* Neang, Chan & Poyarkov, 2018, *S. ouboteri* Pham, Pham, Le, Ngoc, Ziegler & Nguyen, 2024, *S. rufocaudata* (Darevsky and Nguyen, 1983), and *S. rupicola* (Smith, 1916) [7,12,41,47,53]. For other congeners in which toes are in contact with fingers or not when limbs are adpressed, *Scincella chengduensis* **sp. nov.** can be distinguished from *S. baraensis* Nguyen, Nguyen, Nguyen & Murphy, 2020, *S. darevskii* Nguyen, Ananjeva, Orlov, Rybaltovsky & Böhme, 2010, *S. ochracea* (Bourret, 1937), and *S. vandenburghi* (Schmidt, 1927) by having fewer DBR (4 vs. 6–8) [8,9,54,57]. The new species can also be distinguished from *S. boettgeri* (Van Denburgh, 1912), *S. capitanea* Ouboter, 1986, *S. devorator* (Darevsky, Orlov & Cuc, 2004), *S. dunan* Koizumi, Ota & Hikida, 2022, *S. punctatolineata* Boulenger, 1983, and *S. victoriana* (Shreve, 1940) by having fewer T4S (10–12 vs. 12–20) [1,3,5,13,45,55,58,59].

## 4. Discussion

**Urban cryptic biodiversity:** Chengdu is a significant center for economic, scientific, technological, cultural, and transportation activities in southwestern China. Ranking seventh in gross regional product (GDP) nationally and third among sub-provincial cities, Chengdu plays a pivotal role in regional development. Despite this rapid urbanization and economic growth, the city exhibits remarkable biodiversity. The wide altitudinal range, spanning nearly 5000 m (359–5364 m), supports diverse habitats and species. Since 2018, several new taxa have been described, including *Gekko cib* (Reptilia), *Oreolalax longmenmontis* (Amphibia), *Amolops chaochin* (Amphibia), *Liobagrus chengduensis* (Actinopteri), and *Metiochodes tianfuensis* (Insecta) [21,22,23,24,25]. These findings highlight the cryptic diversity within Chengdu and emphasize the need for detailed assessments of urban environments where species adapt to fragmented and human-modified habitats.

The discovery of *Scincella chengduensis* **sp. nov.** in urban Chengdu also highlights the persistence of cryptic biodiversity within metropolitan environments. Despite the pressures of rapid urbanization, Chengdu functions as a biodiversity reservoir, offering refuge for various species [60]. However, an unresolved record of *Scincella tsinlingensis* from urban Chengdu raises questions. Qing et al. (2013) [26] reconstructed a phylogenetic tree that included an individual of ‘*S. tsinlingensis*’, reportedly collected from “Sichuan University campus”, a central urban area of Chengdu. As *S. tsinlingensis* is a palearctic species unlikely to inhabit the Chengdu Plain, this report may represent a misidentification of *Scincella chengduensis* **sp. nov.** Despite extensive field surveys at the campus and surrounding areas, no additional *Scincella* specimens have been located. As the original data from Qing et al. (2013) cannot be verified (Qing Ning, personal communication), further targeted surveys are needed to investigate the historical presence or potential relict populations of *Scincella chengduensis* **sp. nov.** within the same location.

Cryptic biodiversity in urban environments is often overlooked due to several factors. Rapid urbanization and infrastructure development foster the perception that cities are unsuitable for wildlife, reinforced by habitat fragmentation, pollution, and invasive species [61,62]. Infrastructure such as roads and buildings further fragment habitats, reducing the viability of native populations [63]. Species with cryptic behaviors or low population densities, including *Scincella chengduensis* **sp. nov.**, may evade detection without targeted surveys. Research priorities that focus on pristine environments may exacerbate this issue [64,65]. Conservation efforts frequently neglect urban areas, viewing cities as degraded spaces unsuitable for meaningful wildlife preservation. However, cities like Chengdu demonstrate that urban landscapes can sustain species adapted to human-altered habitats [66,67]. The persistence of biodiversity in such settings underscores the necessity of comprehensive and conservation initiatives targeting urban habitats. Recognizing the ecological value of urban spaces is critical for preserving biodiversity in rapidly urbanizing regions. Conservation strategies must prioritize the remaining natural habitats within urban landscapes to protect these often-overlooked species.

**Unique ventral blotches in *Scincella***. Morphological comparisons revealed that *Scincella chengduensis* **sp. nov.** exhibits significant similarity to *S. potanini*, *S. monticola*, and *S. liangshanensis*. However, *S. chengduensis* can be distinguished from these species by the presence of distinct dark blotches on the ventral surface, a trait absent in other *Scincella* species (Figure 6). As the functional and evolutionary significance of these blotches remains unresolved, further research is required to explore their role and origin. In reptiles, ventral coloration often plays a pivotal role in species differentiation, particularly within the contexts of sexual selection and interspecific recognition. For instance, in the family Agamidae, male throat coloration is frequently associated with sexual selection, driving pronounced interspecific variation [68,69]. The case of *Scincella chengduensis* **sp. nov.** underscores the importance of subtle morphological traits in taxonomic research, especially in closely related species where such traits can provide critical insights into evolutionary divergence and adaptation. While coloration alone may not suffice as the primary basis for species delineation, it can serve as a valuable complementary tool, enhancing the resolution of species boundaries when integrated with other morphological and genetic data. These findings emphasize the importance of comprehensive approaches in reptilian taxonomy, particularly for cryptic or closely related taxa.

**Diversity and identification of *Scincella* species in China.** With the addition of the newly described species, the total number of species in the genus *Scincella* is elevated to 42, with 13 species documented in China [2,70], including *Scincella chengduensis* **sp. nov.**, *S. potanini*, *S. monticola*, *S. tsinlingensis*, *S. liangshanensis*, *S. modesta*, *S. huanrenensis*, *S. reevesii*, *S. barbouri*, *S. doriae*, *S. formosensis*, *S. przewalskii*, and *S. schmidti*. The identification of *Scincella chengduensis* **sp. nov.** highlights the underexplored status of reptile diversity in China, reinforcing the need for systematic field surveys to achieve a comprehensive understanding of the country’s herpetological diversity. This discovery reflects the pivotal role of rigorous taxonomic research in uncovering hidden biodiversity, particularly in regions with high ecological complexity. Recognizing the challenges of species identification in this genus, we provide an updated identification key for the *Scincella* species in China, based on Wang and Zhao (1986) [40].


**Diagnostic key to *Scincella* species in China**


1A Supraoculars 3……………………………………………………………***S. przewalskii***

1B Supraoculars 4……………………………………………………………….…………..**2**

2A Toes and fingers overlap when limbs adpressed……………………………***S. doriae***

2B Toes separated or in contact with fingers when limbs adpressed………………….**3**

3A Upper margin of lateral longitudinal striation relatively straight………….……….**4**

3B Upper margin of lateral longitudinal striation wavy……………………………….**10**

4A Dorsal scale rows between dorsolateral stripes 6……………………………………………………………………………….…***S. wangyuezhaoi***

4B Dorsal scale rows between dorsolateral stripes 4……………………………………………………………………………………….………**5**

5A Presence of dark-colored large blotches on ventral…………………………………………………………***Scincella chengduensis* sp. nov.**

5B Absence of dark-colored large blotches on ventral…………………………………..**6**

6A Number of enlarged, undivided lamellae beneath toe IV 10–13………………………………………………………………………………………………**7**

6B Number of enlarged, undivided lamellae beneath toe IV 13–16…………………………………………………………………………………….………**8**

7A Proportion of tympanum diameter/palpebral disc diameter 1.3–2.4………………………………………………………………………….***S. liangshanensis***

7B Proportion of tympanum diameter/palpebral disc diameter 0.6–1.3……………………………………………………………………………………………**8**

8A Midbody scale-row count 23–24…………………………………………………………………….………….***S. monticola***

8B Midbody scale-row count 24–27…………………………………………………………………………….…….***S. potanini***

9A Infralabials 6……………………………………………………….……***S. huanrenensis***

9B Infralabials 7–8……………………………………………………………***S. tsinlingensis***

10A Dorsal scale rows between dorsolateral stripes 4……………………………………………………………………………………***S. schmidti***

10B Dorsal scale rows between dorsolateral stripes 6 or 8………………………………………………………………………………………..……**11**

11A Postnasal pairs mostly 1……………………………………………………***S. reevesii***

11B Postnasals absent………………………………………………………………………**12**

12A Paravertebral scale-row count 70–79…………………………………………………………………………………***S. barbouri***

12B Paravertebral scale-row count 51–65……………………………………………………………………………………………**13**

13A Relative hind-limb length (hind-limb length/snout-vent length) 0.34–0.39……………………………………………………………………………..***S. formosensis***

13B Relative hind-limb length (hind-limb length/snout-vent length) 0.29–0.33…………………………………………………………………………………***S. modesta***

## 5. Conclusions

This study describes a new species of the genus *Scincella*, *Scincella chengduensis* **sp. nov.**, based on three specimens collected from urban and suburban areas in Chengdu, Sichuan, China. Detailed morphological and genetic analyses confirm its clear distinction from all known congeners, stressing the remarkable yet underappreciated biodiversity of urbanized landscapes. The discovery of this new species emphasizes the ecological significance of fragmented habitats in supporting cryptic biodiversity, even within rapidly developing metropolitan areas. However, as Chengdu continues to experience extensive urban expansion, there is an urgent need for further research to evaluate its conservation status and identify potential threats.

## Figures and Tables

**Figure 1 animals-15-00232-f001:**
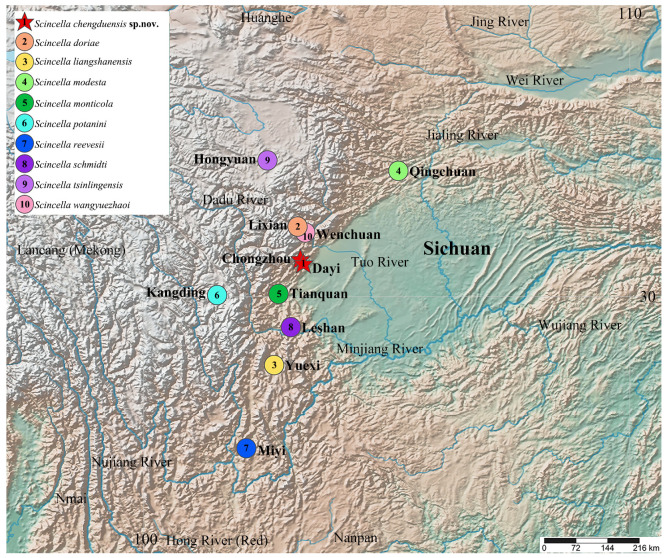
Geographic distribution of *Scincella* species in Sichuan Province, China. 1. *Scincella chengduensis* **sp. nov.** from Chongzhou and Dayi Counties, Chengdu; 2. *S. doriae* from Lixian County; 3. *S. liangshanensis* from Yuexi County; 4. *S. modesta* from Qingchuan County; 5. *S. monticola* from Tianquan County; 6. *S. potanini* from Kangding County; 7. *S. reevesii* from Miyi County; 8. *S. schmidti* from Leshan County; 9. *S. tsinlingensis* from Hongyuan County; 10. *S. wangyuezhaoi* from Wenchuan County. Base maps were obtained from SimpleMapper (https://www.simplemappr.net (accessed on 20 September 2024)).

**Figure 2 animals-15-00232-f002:**
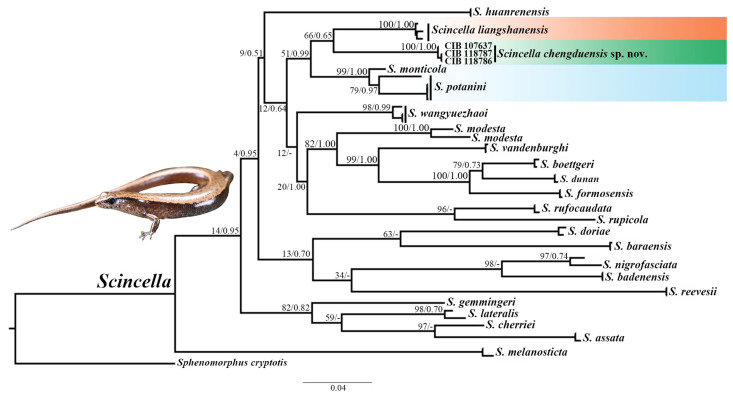
Phylogenetic tree of relationships within the genus *Scincella* reconstructed using four mitochondrial fragments and one nuclear gene. BS from ML analyses and BPP from BI analyses are displayed above branches (BS/BPP). Tip labels correspond to ID numbers listed in Table 1.

**Figure 3 animals-15-00232-f003:**
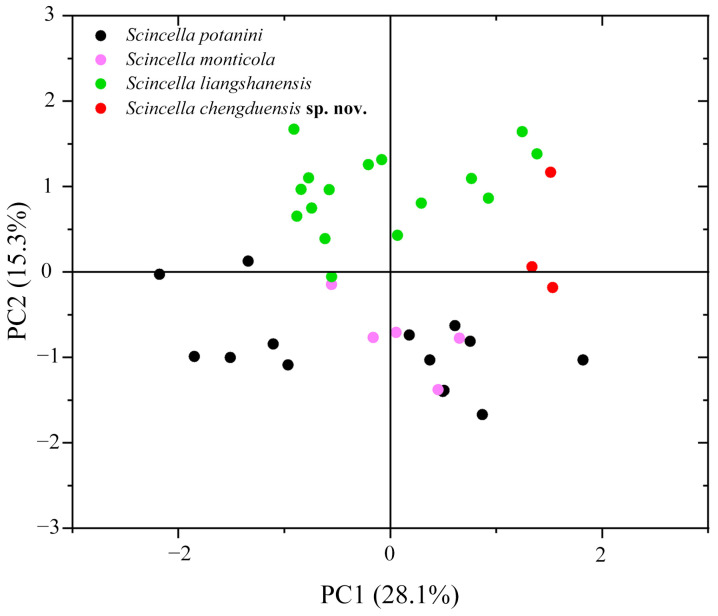
Scatter plot of PC1 and PC2 from PCA based on morphometric measurements, distinguishing the new species and its closely related species. Red, green, black, and light magenta plots represent *Scincella chengduensis* **sp. nov.**, *S. liangshanensis*, *S. potanini*, and *S. monticola*, respectively.

**Figure 4 animals-15-00232-f004:**
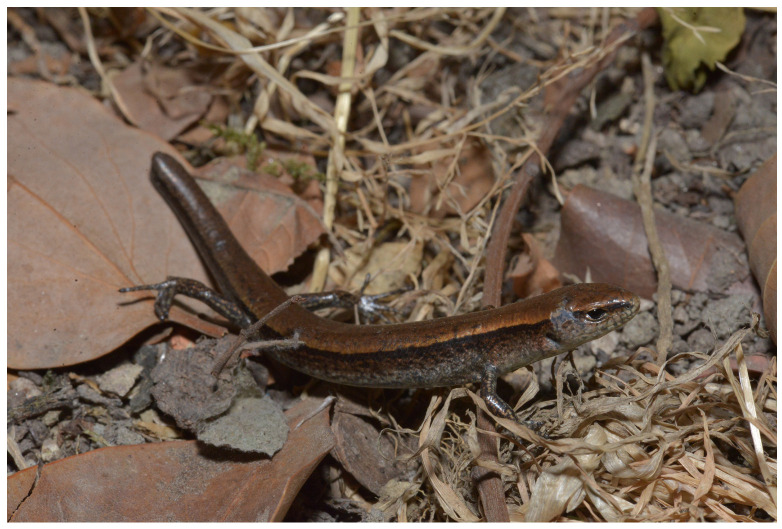
Paratype of *Scincella chengduensis* **sp. nov.** (CIB 118786) in life. Photo by Jin-Long Ren.

**Figure 5 animals-15-00232-f005:**
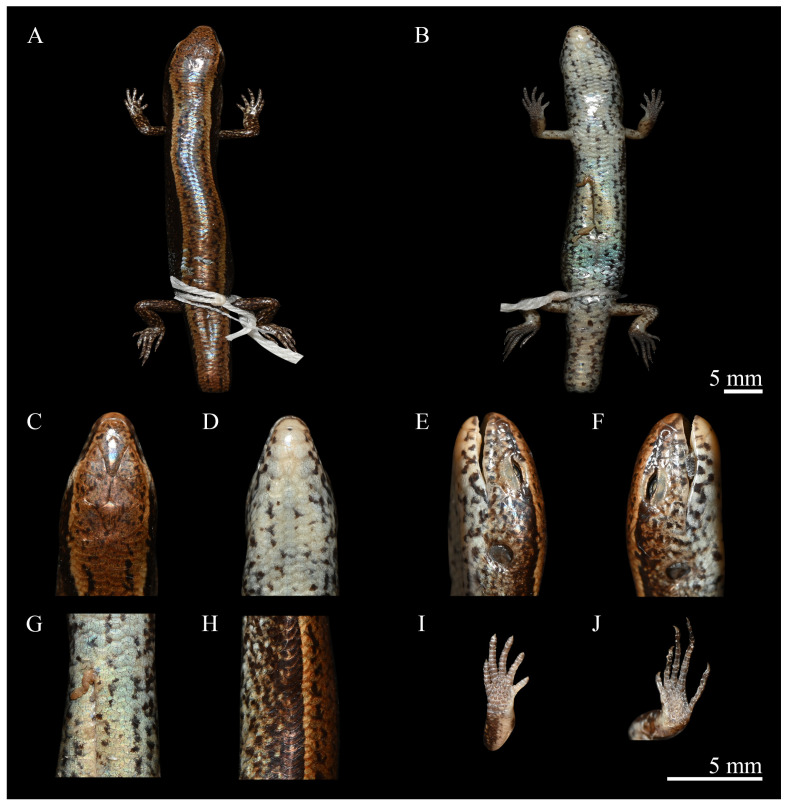
Holotype of *Scincella chengduensis* **sp. nov.** (CIB 118787) in preservative. (**A**): Dorsal view of body; (**B**): Ventral view of body; (**C**): Dorsal view of head; (**D**): Ventral view of head; (**E**): Left view of head; (**F**): Right view of head; (**G**): Ventral feature of body; (**H**): Lateral view of body; (**I**): Ventral view of hand; (**J**): Ventral view of foot. Scale bar: 5 mm. Photos by Zong-Yuan Gao.

**Figure 6 animals-15-00232-f006:**
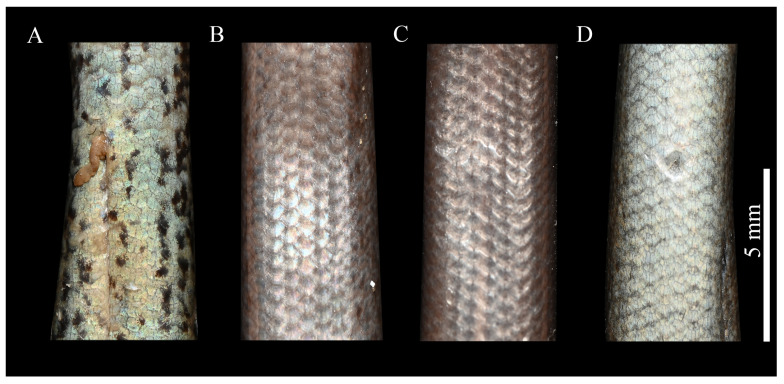
Comparison of ventral features of *Scincella chengduensis* **sp. nov.**, *Scincella liangshanensis*, *S. potanini*, and *S. monticola*. (**A**): *Scincella chengduensis* **sp. nov.**; (**B**): *Scincella liangshanensis*; (**C**): *S. potanini*; (**D**): *S. monticola*. Photographs by Zong-Yuan Gao.

**Figure 7 animals-15-00232-f007:**
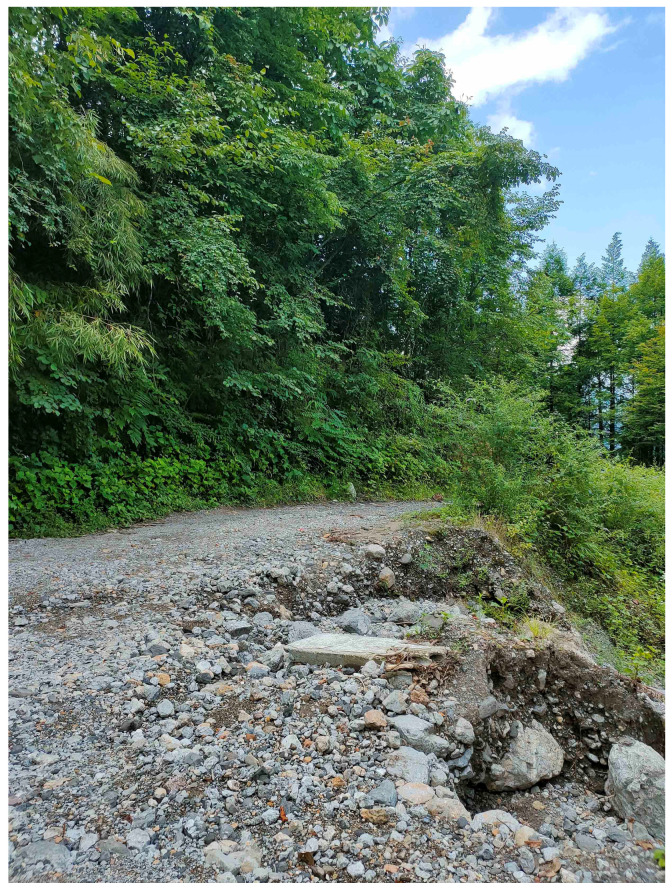
Habitat of *Scincella chengduensis* **sp. nov.** at type locality. Photograph by Jun-Jie Huang.

**Table 1 animals-15-00232-t001:** Localities, voucher information, and GenBank accession numbers for all samples used in this study.

Species	Locality	Voucher ID	16S	12S	*COI*	*Cyt b*	*RAG1*
*Scincella chengduensis* **sp. nov.**	China: Sichuan, Dayi	CIB 107637 ^1^	PQ466921	PQ466924	PQ467109	PQ481143	PQ493651
*Scincella chengduensis* **sp. nov.**	China: Sichuan, Chongzhou	CIB 118786	PQ466920	PQ466923	PQ467108	PQ481144	PQ493652
*Scincella chengduensis* **sp. nov.**	China: Sichuan, Chongzhou	CIB 118787	PQ466919	PQ466922	-	-	-
*Scincella liangshanensis*	China: Sichuan, Yuexi	XM-YXS80 ^1^	PP826313	PP826316	PP824805	–	–
*Scincella liangshanensis*	China: Sichuan, Meigu	CIB 119514	PP826314	PP826318	PP824804	PP849365	–
*Scincella liangshanensis*	China: Sichuan, Meigu	CIB 119513	PP826315	PP826317	PP824806	PP849364	PP849372
*Scincella assata*	El Salvador: Santa Ana, Finca El Milagro	KU 289795 ^1^	JF498074	JF497946	–	–	–
*Scincella assata*	El Salvador: San Miguel, Volcan San Miguel, Canton El Volcan	KU 291286	JF498075	–	–	–	–
*Scincella badenensis*	Vietnam: Tay Ninh, Ba Den Mountain	ITBCZ 5966 ^1^	–	–	MK990602	–	–
*Scincella badenensis*	Vietnam: Tay Ninh, Ba Den Mountain	ITBCZ 5993	–	–	MK990603	–	–
*Scincella baraensis*	Vietnam: Binh Phuoc, Ba Ra Mountain	ITBCZ 6534	–	–	MT742256	–	–
*Scincella baraensis*	Vietnam: Binh Phuoc, Ba Ra Mountain	ITBCZ 6536	–	–	MT742258	–	–
*Scincella boettgeri*	Japan: Southern-Ryukyu Islands, Yaeyama Group	KUZ R68001 ^1^	–	–	LC630768	AB818747	–
*Scincella boettgeri*	Japan: Southern-Ryukyu Islands, Yaeyama Group	KUZ R68008	–	–	LC630770	AB818772	–
*Scincella cherriei*	Mexico: Chiapas, Montes Azules Biosphere Reserve	RCMX235	MW265932	–	–	–	
*Scincella doriae*	Vietnam: Lam Dong, Bidoup-Nui Ba N. P.	ZMMU R-13268-01062 ^1^	–	–	MH119617	–	–
*Scincella doriae*	Vietnam: Lam Dong, Bidoup-Nui Ba N. P.	ZMMU R-13268-00412	–	–	MH119616	–	–
*Scincella dunan*	Japan: Southern Ryukyus, Yonagunijima Is.	KUZ R65170	–	–	LC630778	AB818791	–
*Scincella dunan*	Japan: Southern Ryukyus, Yonagunijima Is.	KUZ R67027	–	–	LC630779	AB818794	–
*Scincella formosensis*	China: Taiwan	KUZ R37516	–	–	LC630790	AB818818	–
*Scincella formosensis*	China: Taiwan	KUZ R37515	–	–	LC630789	AB818814	–
*Scincella gemmingeri*	–	–	AY308294	AY308445	–	–	–
*Scincella huanrenensis*	Korea: Gangwon-do, Pyeongchang-gun	–	NC030779	NC030779	NC030779	NC030779	–
*Scincella huanrenensis*	Korea: Gangwon-do, Pyeongchang-gun	G390SH	KU507306	KU507306	KU507306	KU507306	–
*Scincella lateralis*	–	KU 289460	JF498077	JF497948	–	–	–
*Scincella lateralis*	USA: Texas	DCC 2842 ^1^	HM852503	HM852476	–	–	–
*Scincella melanosticta*	Vietnam: Gia Lai, Kon Chu Rang N.R.	ZMMU NAP-06376	–	–	MH119622	–	–
*Scincella melanosticta*	Vietnam: Gia Lai, Kon Chu Rang N.R.	ZMMU NAP-05519	–	–	MH119621	–	–
*Scincella modesta*	China: Zhejiang, Ningpo	WYF 11520 ^1^	–	PP819197	PP819215	PP849366	PP849370
*Scincella modesta*	China: Zhejiang, Ningpo	CIB 121415	PP819195	PP819198	PP819217	–	–
*Scincella monticola*	China: Yunnan, Shangri-La	DL-YNJC2020824 ^1^	OP955962	OP955952	–	–	–
*Scincella nigrofasciata*	Vietnam: Ba Ria-Vung Tau, Dinh Mountain	ITBCZ 6344 ^1^	–	–	MK990605	–	–
*Scincella nigrofasciata*	Cambodia: Mondulkiri, Keo Seima W.S.	CBC 02546 ^1^	–	–	MH119614	–	–
*Scincella potanini*	China: Sichuan, Kangding	DL-KD202109071	OP935937	OP942203	OP942210	OQ448540	PP849368
*Scincella potanini*	China: Sichuan, Kangding	DL-KD202109072	OP935987	OP942208	OP942209	PP849367	PP849369
*Scincella potanini*	China: Sichuan, Kangding	DL-KD2018070302	OP935989	OP942204	OP942211	OQ448542	–
*Scincella potanini*	China: Sichuan, Kangding	XM6920	OP935998	OP942205	OP942212	OQ448543	PP849371
*Scincella reevesii*	China: Guangdong, Zhaoqing	NB2017030715	NC054206	NC054206	NC054206	–	–
*Scincella reevesii*	China: Guangdong, Zhaoqing	–	MN832615	MN832615	MN832615	–	–
*Scincella rufocaudata*	Vietnam: Ha Tinh	ZFMK 76239 ^1^	HM773217	–	–	–	–
*Scincella rufocaudata*	Vietnam: Ha Tinh	ZFMK 76238	HM773216	–	–	–	–
*Scincella rupicola*	Thailand	KUZ 40458	AB057403	AB057388	–	–	–
*Scincella vandenburghi*	Japan: Tsushima Is.	KUZ R66394	–	–	LC507695	LC507740	–
*Scincella vandenburghi*	Korea: Yeongwol-gun	G389SV	KU646826	KU646826	KU646826	KU646826	–
*Scincella wangyuezhaoi*	China: Sichuan, Wenchuan	CIB 87246	OP941172	OP942191	OQ402205	–	–
*Scincella wangyuezhaoi*	China: Sichuan, Lixian	CIB 119518	OP941173	OP942193	–	–	–
*Scincella wangyuezhaoi*	China: Sichuan, Lixian	CIB 119510	OP941174	OP942192	OQ402206	–	–
*Sphenomorphus cryptotis*	China: Guangxi, Shangsi	CIB 119027	OP942190	OP942206	OP942215	OQ448544	–

^1^ All abbreviations and catalog numbers are as described in Jia et al. (2024) [15].

**Table 2 animals-15-00232-t002:** Morphometric and meristic traits of *Scincella chengduensis* **sp. nov.** (provided in mm). Morphological character abbreviations are detailed in the Materials and Methods section.

	Holotype	Paratypes		Range
Specimen Number	CIB 118787	CIB 118786	CIB 107637	
sex	Male	Female	Juvenile	-
SVL	37.7	43.2	28.4	28.4–43.2
TaL	59.9	-	-	-
original tail	Yes	*	*	-
TaW/SVL	0.11	0.10	0.10	0.10–0.11
TaD/SVL	0.10	0.09	0.06	0.06–0.10
HL/SVL	0.18	0.18	0.20	0.18–0.20
HW/SVL	0.13	0.13	0.14	0.13–0.14
HD/SVL	0.11	0.10	0.10	0.10–0.11
ED/SVL	0.05	0.06	0.05	0.05–0.06
PDD/SVL	0.02	0.02	0.03	0.02–0.03
TD/PDD	1.58	1.05	1.07	1.05–1.58
TD/SVL	0.03	0.02	0.03	0.02–0.03
END/SVL	0.04	0.03	0.05	0.03–0.05
SNL/SVL	0.07	0.06	0.07	0.06–0.07
IND/SVL	0.04	0.05	0.04	0.04–0.05
IOD/SVL	0.08	0.08	0.08	0.08
MBW/SVL	0.13	0.15	0.21	0.13–0.21
MBD/SVL	0.12	0.09	0.11	0.09–0.12
AGD/SVL	0.61	0.55	0.56	0.55–0.61
FLL/SVL	0.26	0.24	0.21	0.21–0.26
HLL/SVL	0.32	0.27	0.23	0.23–0.32
T4L/SVL	0.10	0.09	0.08	0.08–0.10
F4L/SVL	0.06	0.06	0.05	0.05–0.06
MBSR	23	23	23	23
PVSR	60	57	60	57–60
VSR	44	42	43	42–44
Gulars	21	22	21	21–22
DBR	4	4	4	4
AGSR	58	52	56	52–58
F4S left/right	9/9	8/9	9/9	8–9/9
T4S left/right	12/12	10/10	11/11	10–12/10–12
NU left/right	3/3	3/3	4/3	3–4/3
SL left/right	7/7	7/7	7/7	7/7
IfL left/right	7/7	7/6	7/7	7/6–7
SC left/right	6/6	7/7	6/6	6–7/6–7
SO left/right	4/4	4/4	4/4	4/4
TEM left/right	1 + 2/1 + 2	2/2	2/2	1 + 2 or 2
L left/right	2/2	2/2	2/2	2/2
Chin-shields (pair)	3	3	3	3
FTSR	2	2	2	2
MT left	20	18	18	18–20
LT left	20	18	16	16–20
PF	No	Yes	Yes	Yes or No
FP	Yes	Yes	Yes	Yes
P	Yes	Yes	Yes	Yes
DLBV	Presence	Presence	Presence	Presence
Limbs adpressed	No	No	No	No

* Tail incomplete.

**Table 3 animals-15-00232-t003:** Uncorrected *p*-distances (%) for 16S rRNA sequences of *Scincella* species analyzed in this study.

	Taxa	GenBank No.	1	2	3	4	5	6	7	8	9	10	11	12	13	14	15	16	17
1	*Scincella chengduensis* **sp. nov.**	PQ466921																	
2	*Scincella chengduensis* **sp. nov.**	PQ466920	0.0																
3	*Scincella chengduensis* **sp. nov.**	PQ466919	0.0	0.0															
4	*Scincella liangshanensis*	PP826313	3.0	3.0	3.0														
5	*Scincella monticola*	OP955962	4.8	4.8	4.8	5.3													
6	*Scincella potanini*	OP935937	5.5	5.5	5.5	6.9	3.2												
7	*Scincella wangyuezhaoi*	OP941172	5.5	5.5	5.5	6.4	7.9	8.3											
8	*Scincella modesta*	PP819195	6.0	6.0	6.0	7.2	7.2	7.9	7.9										
9	*Scincella vandenburghi*	KU646826	7.2	7.2	7.2	8.6	7.5	10.1	8.1	7.4									
10	*Scincella huanrenensis*	NC030779	7.9	7.9	7.9	9.1	7.7	8.6	8.8	11.4	8.7								
11	*Scincella lateralis*	JF498077	8.0	8.0	8.0	8.7	8.2	9.7	9.5	7.8	8.3	11.4							
12	*Scincella gemmingeri*	AY308294	8.2	8.2	8.2	8.2	9.2	9.2	9.7	10.2	12.3	11.1	8.7						
13	*Scincella assata*	JF498074	8.7	8.7	8.7	9.7	9.7	9.2	10.5	9.7	11.1	9.2	9.1	8.4					
14	*Scincella cherriei*	MW265932	8.9	8.9	8.9	8.3	9.1	10.5	10.7	10.5	11.1	8.9	7.9	8.1	6.1				
15	*Scincella rupicola*	AB057403	9.5	9.5	9.5	11.2	10.0	10.7	10.8	9.5	10.1	11.1	9.6	11.0	11.6	12.0			
16	*Scincella rufocaudata*	HM773217	9.8	9.8	9.8	11.6	10.6	10.6	11.1	9.1	9.7	11.1	7.7	12.0	12.0	11.3	7.5		
17	*Scincella reevesii*	NC054206	10.4	10.4	10.4	10.2	11.2	10.9	10.9	11.0	12.6	13.3	11.0	13.1	15.0	15.5	12.3	12.9	

**Table 4 animals-15-00232-t004:** Diagnostic morphometric comparison between *Scincella chengduensis* **sp. nov.** and three morphologically similar congeners from Southwest China. Morphological character abbreviations are detailed in the Materials and Methods section.

Selected Characters	*Scincella chengduensis* sp. nov.	*Scincella liangshanensis*	*S. potanini*	*S. monticola*
	*N* = 3	*N* = 16	*N* = 14	*N* = 4
SVL	28.4–43.2	43.1–61.9	26.6–57.9	36.3–53.0
TaL/SVL	1.59	0.96–1.71	1.02–1.12	1.61
TaW/SVL	0.10–0.11	0.08–0.11	0.07–0.11	0.07–0.10
TaD/SVL	0.06–0.10	0.08–0.10	0.07–0.10	0.08–0.10
HL/SVL	0.18–0.20	0.13–0.19	0.12–0.20	0.16–0.17
HW/SVL	0.13–0.14	0.11–0.15	0.09–0.16	0.13
HD/SVL	0.10–0.11	0.09–0.11	0.08–0.12	0.10–0.11
ED/SVL	0.05–0.06	0.03–0.05	0.03–0.05	0.04–0.05
PDD/SVL	0.02–0.03	0.01–0.03	0.01–0.03	0.02
TD/SVL	0.02–0.03	0.02–0.03	0.01–0.03	0.01–0.02
TD/PDD	1.05–1.58	1.25–2.38	0.79–1.25	0.62–1.11
END/SVL	0.03–0.05	0.04–0.05	0.03–0.05	0.04–0.05
SNL/SVL	0.06–0.07	0.06–0.08	0.05–0.08	0.06–0.07
IND/SVL	0.04–0.05	0.03–0.05	0.03–0.05	0.04–0.05
IOD/SVL	0.08	0.06–0.09	0.06–0.10	0.07–0.08
AGD/SVL	0.55–0.61	0.56–0.66	0.52–0.72	0.56–0.65
MBW/SVL	0.13–0.21	0.12–0.18	0.09–0.20	0.11–0.16
MBD/SVL	0.09–0.12	0.10–0.15	0.09–0.13	0.10–0.16
FLL/SVL	0.21–0.26	0.14–0.22	0.11–0.25	0.13–0.19
HLL/SVL	0.23–0.32	0.22–0.33	0.17–0.31	0.20–0.22
F4L/SVL	0.05–0.06	0.05–0.07	0.03–0.06	0.03–0.05
T4L/SVL	0.08–0.10	0.07–0.13	0.05–0.10	0.05–0.07
MBSR	23	23–27	24–27	23–25
DBR	4	4	4	4
SRB	1–2.5	1.5–2.5	1.5–3	1.5–2
PVSR	**57–60**	69–80	62–80	69–73
VSR	**42–44**	43–57	45–64	45–52
Gulars	21–22	22–29	23–25	22–24
VSR+gulars	**64** **–** **65**	68–82	69–89	67–77
AGSR	**52–58**	60–79	61–82	65–74
F4S left	8–9	8–11	7–10	8–10
T4S left	10–12	10–15	10–13	10–12
MT left	18–20	15–22	9–20	14–20
LT left	16–20	16–21	10–18	12–19
NU left	3–4	2–5	3	3–4
SL left	7	7	7–8	7
IfL left	7	6–7	6–7	6–7
SC left	6–7	6–7	6–7	6–7
SO left	4	4	4	4
TEM left	1 + 2–2	1 + 2–2 + 3	1 + 2–2 + 2	1 + 2–2 + 2
Chin-shields (pair)	3	3	3	3
FTSR	2	2	2	2
PF	Yes	Yes/No	Yes/No	Yes
FP	Yes	Yes	Yes	Yes
P	Yes	Yes/No	Yes	Yes
DLBV	**Presence**	Absence	Absence	Absence

## Data Availability

All necessary details of the material described, including locations, dates and the name of the collector, are available in this article. Upon reasonable request, the material can be made available by the author.

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
