# Peer review of "Hidden Urban Biodiversity: A New Species of the Genus Scincella Mittleman, 1950 (Squamata: Scincidae) from Chengdu, Sichuan Province, Southwest China"

_animals, 2025, doi:10.3390/ani15020232_

Round 1

Reviewer 1 Report

Comments and Suggestions for Authors

The manuscript describes a new species of the genus Scincella. However, the evidence of the supposed distinctiveness is not adequate. I strongly suggest to include also osteological characters in the differentiation and especially to provide a Figure of the cranium, including also the lower jaw, of the new taxon. This can be done via microCT scanning or even dried skeletal specimen. A brief description of the skull should also be included, ideally also to figure some skull of some other closely related species. Without these, the manuscript I fear would not be strong enough to be published (and actually so it should not be published). I therefore recommend major revision.

Regards

Reviewer 2 Report

Comments and Suggestions for Authors

I have carefully read the manuscript where the authors describe a new species of Scincella from Sichuan Province, Chengdu city, based on morphological differences and significant phylogenetic distances in the analysis of foure mitochondrial and one nuclear gene fragments. The work is done in a classical style with observance of all standard aspects of taxonomical description of a new species. The research material is neatly and logically presented. The validity of the new species is not in doubt.

However, I have a few comments: 

1. Regarding the volume of the studied material, it is not clear to me how many individuals were processed and used in the morphological analysis, how many new sequences were obtained directly in this work and how many were taken from the genebank. This should be clarified and make it clear for the reader. In addition, I recommend using and uploading the primary measurement data to Morphobank so that the reader can familiarize themselves with the data used.

2. The limitation of the type series to only three specimens is not clear. Judging from the text of the paper, the authors had a rather large comparative material on the new species, but only three specimens were included in the type series. If the authors do not have additional material that could be added to the type series, it should be indicated in the text of the manuscript. Then it will become clear why only 3 specimens of the new species were used in the PCA analysis. If the authors have possibility to add more samples to the type series, it would be good that type series will consist of 5 males and 5 females of the new species.

And we need some explanation and disscussion why one adult male, one adult female and juv specimens where used in PCA.

3. it is necessary to add a photo of the whole type series (dorsal and ventral view). 

4. Since the species was found in an urbanized area, it would be logical question from reader about the natural origin of this population. After all, the species could be introduced there by accident.  Will be great if the authors can add one short paragraph in disscussion section regarding this issue. 

Other minor remarks can be found in the text.

I definitely recommend this manuscript for publication after some minor revision and clarification.

Reviewer 3 Report

Comments and Suggestions for Authors

see attached file

Round 2

Reviewer 1 Report

Comments and Suggestions for Authors

I still have the same opinion that the cranial anatomy of the new species should be figured and described.

Author Response

Thank you for taking the time to review this manuscript. We appreciate your comment and feedback. Below, we have provided a detailed response to the point raised. Please see the file attached.
